# Towards Explainable Depression Detection: A Neurosymbolic Approach to Uncover Social Media Signals with Generative AI

**Mohammad Saeid Mahdavinejad**                              SAEID@KSU.EDU
*Kansas State University*

**Peyman Adibi**                              ADIBI@ENG.UI.AC.IR
*University of Isfahan*

**Amirhassan Monadjemi**                              AMIR@COMP.NUS.EDU.SG
*National University of Singapore*

**Pascal Hitzler**                              HITZLER@KSU.EDU
*Kansas State University*

**Editors:** Leilani H. Gilpin, Eleonora Giunchiglia, Pascal Hitzler, and Emile van Krieken

## Abstract

Depression remains a pervasive mental health disorder that demands prompt diagnosis and intervention. Although social media data presents a promising avenue for early detection, traditional deep neural models are frequently critiqued for their lack of interpretability and susceptibility to bias. We introduce ProtoDep[1]—a neurosymbolic framework that integrates clinically grounded categorizations (e.g., PHQ-9 symptoms) with large language model–assisted prototype learning. Unlike conventional black-box models, ProtoDep aligns individual tweets with symptom-level prototypes, offering interpretable explanations at three levels: (i) symptom-level insights that map user posts to recognized depressive patterns, (ii) case-based reasoning that compares users to representative prototype profiles, and (iii) transparent concept-level decisions, wherein classification at inference time is driven by the distances between the user profile and prototype user and symptom clusters, yielding clear, quantifiable explanations. By integrating symbolic mental health constructs with neural embeddings, ProtoDep achieves a mean F1-score of 94% across five benchmark datasets and establishes a foundation for interpretable depression screening pipelines with potential applicability in clinical settings.

## 1. Introduction

Depression is a prevalent mental health disorder affecting millions worldwide. According to the National Institute of Mental Health, approximately 22.8% of adults in the U.S. experience a diagnosable mental illness annually of Mental Health (2023). Timely diagnosis and intervention are crucial, as untreated depression can lead to severe consequences, including suicide and chronic, risky behaviors such as substance abuse Goodwin et al. (2022). Traditional methods, such as self-reported questionnaires or clinical interviews, often suffer from low participation and potential selection biases, prompting the exploration of social media as a source of real-time insights into users' mental health Chancellor and De Choudhury

---

1. Code and supplementary material are available at `https://github.com/MahdaviNejad/ProtoDep`

(2020); Culotta (2014); De Choudhury and De (2014); Guntuku et al. (2017); Paul and Dredze (2011); Yazdavar et al. (2020).

Although deep learning has shown promise in extracting latent signals of depression from online text, its opaque, "black box" nature hinders interpretability Ji et al. (2021); Nguyen et al. (2022). This lack of transparency raises concerns about potential biases and errors, particularly in sensitive domains like mental health. In response, researchers have proposed neurosymbolic frameworks that combine the representational power of neural networks with the clarity of symbolic knowledge. In mental health contexts, clinically grounded categories such as PHQ-9 can anchor data-driven features in an expert-derived taxonomy, bridging low-level text signals with higher-level conceptual definitions Nguyen et al. (2021); Han et al. (2022); Bibal et al. (2022).

We introduce **ProtoDep**, a novel neurosymbolic framework designed to make depression detection on social media both accurate and interpretable. ProtoDep integrates mental health knowledge (e.g., PHQ-9 symptom categories) with neural prototype learning guided by large language models (LLMs), enabling multi-level explanations: symptom-level interpretation, case-based reasoning, and transparent feature weighting.

To illustrate *how* ProtoDep captures these explanatory dimensions, let $U$ be the set of users, $T$ the set of tweets, and $S$ the set of depressive symptoms. We define two core relations, authoredBy $\subseteq T \times U$ to capture which user wrote each post, and exhibitsSymptom $\subseteq (T \cup U) \times S$ to indicate symptom manifestations in either a text or a user. From a *symptom-level* perspective, if a user $u \in U$ exhibits symptom $s \in S$ through at least one authored post $t \in T$, we have:

$$\exists t \in T \quad \big(\text{authoredBy}(t, u) \wedge \text{exhibitsSymptom}(t, s)\big) \;\Rightarrow\; \text{exhibitsSymptom}(u, s). \quad (1)$$

This logical rule captures how local indicators of depressive behavior (found in individual posts) propagate to a user-level symptom label. For *case-based reasoning*, let similarTo $\subseteq U \times U$ encode user-to-user similarity, and define $C_u = \{\, c \in U \mid \text{similarTo}(u, c)\}$. If a similar user $c \in C_u$ exhibits symptom $s$, then:

$$\exists c \in C_u \quad \big(\text{similarTo}(u, c) \wedge \text{exhibitsSymptom}(c, s)\big) \;\Rightarrow\; \text{support}(u, s). \quad (2)$$

Hence, ProtoDep can support a symptom in user $u$ by drawing analogies with similar users. Finally, *transparent decision-making* arises from a weighted function over features $F = \{f_1, f_2, \dots\}$ with corresponding weights $W = \{w_1, w_2, \dots\}$. Let $\Psi : (F \times W) \to D$ be a decision function mapping these features into a final label $D \in \{\text{Depressed}, \text{NotDepressed}\}$. For each user $u \in U$,

$$\text{decision}(u) \;=\; \Psi\big(\{(f_i, w_i) \mid f_i \in F_u\}\big), \quad (3)$$

ensuring that each classification for $u$ is grounded in a transparent, feature-based rationale. This multi-level reasoning—spanning symptom cues, case-based analogies, and explicit feature weights—makes ProtoDep both powerful and interpretable for depression detection on social media. While our evaluation relies on synthetic symptom labels, the direct mapping of learned prototypes to PHQ-9 categories demonstrates clear alignment with established clinical criteria, suggesting ProtoDep could serve as a first-pass screening aid by providing clinicians with interpretable, symptom-level flags for follow-up assessment.

## 2. Related Work

Leveraging social media data for mental health analysis offers unique advantages over conventional clinical surveys, including its real-time nature and large-scale user engagement. Early research focused on dictionary-based methods and straightforward linguistic features to detect depressive symptoms on Twitter and related platforms Yazdavar et al. (2017); Culotta (2014); De Choudhury and De (2014); Paul and Dredze (2011), laying the groundwork for subsequent efforts integrating richer textual, behavioral, and network signals Chancellor and De Choudhury (2020); Guntuku et al. (2017); Birnbaum et al. (2017). Researchers have since broadened the scope to multiple mental health conditions—encompassing anxiety, suicidality, and mood instability—via feature engineering and machine learning pipelines Ahmed et al. (2022); Saifullah et al. (2021); Shen and Rudzicz (2017); Burnap et al. (2015); Coppersmith et al. (2018); Saha et al. (2017).

Deep neural architectures later emerged as a dominant solution for mental health detection, owing to their capacity to learn nuanced language representations directly from data Shen et al. (2017); De Choudhury et al. (2013a,b); Tadesse et al. (2019). Pre-trained language models, such as BERT variants, demonstrated marked gains in depression detection Han et al. (2022); Ji et al. (2021); Nguyen et al. (2022), while multi-task learning (MTL) enabled modeling of multiple mental health conditions simultaneously Benton et al. (2017); Sarkar et al. (2022). Yet these neural approaches often behave as black boxes, yielding minimal insight into *why* certain features or posts are linked to depression. Post-hoc attribution tools, including LIME and SHAP, generate feature importance measures but rarely offer contextually grounded rationales crucial for clinical interpretation Ribeiro et al. (2016); Shapley et al. (1953); Lundberg and Lee (2017).

To address this challenge, *neurosymbolic* strategies embed symbolic knowledge within neural architectures to enhance explainability and adaptability Khandelwal et al. (2024); Dalal et al. (2024). In mental health research, such methods can leverage domain-specific resources, for example SenticNet for emotional concept grounding Dou and Kang (2024), or employ knowledge graphs to capture clinically validated constructs Khandelwal et al. (2024); Dalal et al. (2024). Additionally, prototype-based explainable models provide case-based reasoning at the concept level, enabling predictions to be traced to exemplars Das et al. (2022); Ni et al. (2022); Zhang et al. (2021a). However, most previous work fails to unite symptom-level clarity, user-level comparisons, and robust adaptation without expensive annotated datasets.

We build on Yazdavar et al. Yazdavar et al. (2017)'s semi-supervised alignment of social media signals with clinical symptom lexicons, adopting their expert-curated categories as a silver-standard proxy in lieu of new clinician annotations.

*ProtoDep* bridges these gaps by fusing symbolic mental health constructs (e.g., PHQ-9 symptoms) with learned neural prototypes, producing multi-level, meaningful explanations —symptom-level alignments, user-level case comparisons, and transparent classification weights. Our PRIDE analysis confirms that learned prototypes are distinct and coherent, establishing a foundation for interpretable depression screening with potential applicability in clinical settings, pending dedicated expert evaluation.

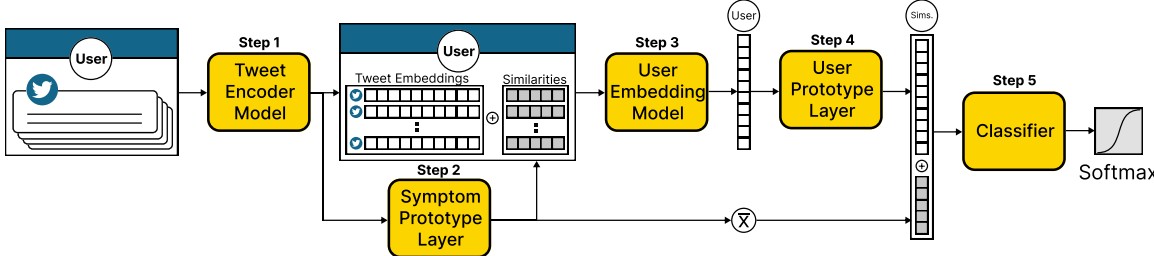

Figure 1: Overview of ProtoDep: (1) tweet encoding, (2) PHQ-9–aligned symptom proto-
types, (3)symptom-aware user embedding, (4) user prototype learning, and (5)
transparent classification.

## 3. Method

ProtoDep uses prototype learning to represent symptoms and user behaviors, providing
clearer explanations for its predictions. The framework comprises five steps, illustrated in
Figure 1. **Step 1**: Embedding user tweets, **Step 2:** Learning symptom prototypes, **Step
3:** Encoding user behaviors, **Step 4:** Learning user prototypes, and **Step 5:** Performing
classification.

### 3.1. Preliminaries

Let $U$ be the set of users and $T$ the set of all tweets. The relation authoredBy $\subseteq T \times U$
encodes which user wrote each post. For each user $u \in U$, define

$$T(u) = \{\, t \in T \mid \mathsf{authoredBy}(t, u)\},$$

i.e., the set of posts authored by $u$. Our goal is to predict a binary label $\hat{y} \in \{0, 1\}$ for each
user $u$, where $\hat{y} = 1$ indicates depression and $\hat{y} = 0$ indicates non-depression.

### 3.2. Step 1: Embedding User's Tweets

Given a user $u \in U$, let $T(u)$ be the set of tweets authored by $u$. We first obtain an
embedding matrix $E_u$ by applying a pre-trained sentence encoder to each text in $T(u)$:

$$E_u = \text{TweetEncoder}\big(T(u)\big). \tag{4}$$

The TweetEncoder model has the potential to significantly affect the quality of the learned
prototypes. Section B.5 will offer more details on this.

### 3.3. Step 2: Learning Symptom Prototypes

This step focuses on training symptom prototypes that faithfully capture the essence of
each depression symptom while maintaining close alignment with actual tweets from the
dataset. In simpler terms, we aim to develop representations of symptoms that are both
accurate and grounded in the language used by individuals describing their experiences.
However, due to the absence of information about individual user symptoms or specific

tweets in the dataset (limited to user-level labels), we propose a supervised initialization strategy for prototypes, coupled with a specific loss function, to achieve this objective.

**A. Symptom Space Creation:** The first step in creating an embedding space for symptom prototypes is identifying the underlying concepts or symptoms within the space. We use the Patient Health Questionnaire - 9 (PHQ-9[2]), one of the most widely used questionnaires to assess depression Kroenke et al. (2010), as a reference for defining the concepts. The PHQ-9 is a self-administered questionnaire that measures the presence and severity of nine depressive symptoms over two weeks. It has been validated by multiple studies and is regarded as a reliable and accurate measure of depression Levis et al. (2019). The second step is to initialize a set of prototypes for each concept. Manually creating exemplary sets for each prototype proves resource-intensive and needs iterative refinement. To address this challenge, we leverage the generative capabilities of LLMs. Specifically, we employ GPT-4 to automatically generate relevant examples, focusing on different aspects of a given symptom. These examples serve as our initial set of prototypes for subsequent training. We note that the number of prototypes is an important hyperparameter, and generating different numbers of examples from GPT-4 for each experiment is inconsistent and impractical. Therefore, we generate a maximum number of examples once and use the mean of the embedded examples as a base prototype for each symptom. Then, for each experiment, we sample around each base prototype with a normal distribution. We define $p_{base}^{j}$ as the base prototype and $P^j$ to be set of $m$ prototypes for the symptom $j$ by:

$$P^j \sim \mathcal{N}(p_{base}^{j}, \sigma^2).$$ (5)

where $\sigma^2$ is variance. Therefore, $P$ will be set for all symptom prototypes. For reproducibility, the exact GPT-4 prompts used to generate these examples appear in Section B.8 of the appendix, and Figure 5(a) provides a visual overview of this step.

**B. Symptom Space Optimization:** Given the lack of tweet-level labels, we propose a novel approach that leverages supervised-initialized prototypes. Specifically, we formulate the optimization process as a multi-label classification task, where each tweet is labeled with the nearest symptom within the embedding space. By adopting this strategy, we effectively use prior knowledge from the initial symptom space while accommodating the lack of labeled tweets. As a result, we define the total symptom loss as the sum of two terms:

$$L_{symp} = \lambda_1 * L_{sinkhorn} + \lambda_2 * L_{mse}$$ (6)

where $\lambda_1$ and $\lambda_2$ are hyperparameters, and $L_{sinkhorn}$ is Sinkhorn loss, a mathematical tool that computes optimal transport between two probability distributions Cuturi (2013). This choice has several advantages over conventional loss functions. It enhances the stability and robustness of the training process, mitigates the impact of noise and overfitting, and accelerates the convergence rate Feydy et al. (2019). For calculating $L_{sinkhorn}$, first, we calculate a cosine similarity between a tweet embedding $e_i$ and a symptom $p_k^j$:

$$\text{sim}_{i,j,k} = \cos(e_i, p_k^j)$$ (7)

where $i \in \{1, ..., n\}$ and $j \in \{1, ..., 9\}$ and $k \in \{1, ..., m\}$. Although cosine similarity may mis-assign borderline tweets, our Sinkhorn regularization promotes cluster cohesion; exploring alternatives (e.g., Wasserstein distances over embedding distributions) remains

---

2. https://www.apa.org/depression-guideline/patient-health-questionnaire.pdf

future work. Then, we assign a label denoted as $s_i$ to each tweet based on its nearest symptom.

$$s_i = \arg\max_j \frac{1}{m} \sum_{k=1}^{m} \text{sim}_{i,j,k} \tag{8}$$

By defining all tweet embeddings with the same symptom as:

$$E^j = \{e_i | s_i = j\} \tag{9}$$

The $L_{sinkhorn}$ will be:

$$L_{sinkhorn} = \frac{1}{9} \sum_{j=1}^{9} \text{sinkhorn}(E^j, P^j) \tag{10}$$

Finally, $L_{mse}$ is a mean squared error loss and a measure of the difference between the input samples and their reconstructions using the nearest prototype. It encourages the prototypes to be representative of the input data. For this purpose, we find the index of the nearest prototype to each tweet embedding $c_i$ as:

$$c_i = \arg\max_{j,k} \text{sims}_{i,j,k} \tag{11}$$

Next, we define the nearest prototype to $e_i$ as:

$$p_{c_i} = \{p_k^j | (j, k) = c_i\} \tag{12}$$

Now the $L_{mse}$ will be calculated as:

$$L_{mse} = \frac{1}{n} \sum_{i=1}^{n} (e_i - p_{c_i})^2 \tag{13}$$

### 3.4. Step 3: Encoding the User

We employ a multi-layer attention mechanism to model user behavior followed by a feedforward neural network. The encoder model for the ProtoDep framework can vary depending on the problem domain and the data modality, which we will elaborate on in section B.3. This step also incorporates similarity scores between tweets and symptom prototypes to enrich user representations:

$$\text{SympSims} = \frac{1}{m} \sum_{k=1}^{m} \text{sim}_{i,j,k} \tag{14}$$

And user embedding $e_u$ will be:

$$e_u = \text{UserEncoder}(E \oplus \text{SympSims}) \tag{15}$$

This composite representation improves the model's understanding of user behavior by highlighting associations with specific symptoms.

### 3.5. Step 4: Learning User Prototypes

This step provides transparent, case-based reasoning to evaluate the user's depressive behavior. It follows the same principle as the learning symptom prototype step and consists of two sub-steps: **A. User Space Creation** and **B. User Space Optimization**. To elucidate this step, we have included Figure 5 (b) in the appendix.

**A. User Space Creation:** Social media datasets for depression detection often exhibit an imbalance between the number of users or tweets in each class. This may negatively impact the reasoning of deep learning models as they may prioritize the majority class during training. Inspired by Das et al. (2022), we encourage the model to find the best examples for both classes to find a more effective decision boundary between them. Unlike the symptom prototype space, which relies on predefined prototypes, the user prototype space allows the model to learn the prototypes from the data. Consequently, we randomly initialize $k$ different vectors per class as initial prototypes.

**B. User Space Optimization:** In this step, we adopt the same optimization strategy as in step 2, but with a crucial difference. We leverage the user-level labels to learn the prototypes in a supervised fashion—this way, we do not require the computation of $s_i$. We denote the total loss for this step as $L_{user}$.

### 3.6. Step 5: Classification

The final step of our model is to classify the users based on their similarity to symptoms and user prototypes. First, we calculate the average of all tweet-symptom similarities, providing an overall measure of the similarity between a user's tweets and the symptom prototypes. Then, we concatenate these scores with the user prototype similarities and feed this into a linear layer followed by a Softmax function to obtain the final classification. We use binary cross-entropy (BCE) loss for this step, and the total loss function for our model will be:

$$L = L_{symp} + L_{user} + L_{BCE} \tag{16}$$

## 4. Experiment Results

We conducted extensive evaluations to assess ProtoDep's performance and interpretability in depression detection. This section presents our findings across three key areas: classification performance, prototype explainability, and transparent decision-making.

**Dataset.** We use the publicly available MDL Twitter (X) dataset for depression screening, following the 60/20/20 train/val/test splits of Han et al. (2022). See Appendix A for user counts, tweet counts, and full labeling details.

**Baselines.** To evaluate the ProtoDep framework, we compare its performance to four established depression detection baselines. Given its similarity in approach, we consider Han et al. (2022) the most relevant and significant baseline for our model. Additionally, we compare our results to Gui et al. (2019); Lin et al. (2020); Zhang et al. (2021b), as these studies appeared to be most pertinent to our work.

**Setup.** We trained all models using a GeForce RTX 3090 with 64GB of RAM and Pytorch 2.0.1. We tuned the hyperparameters on the validation data and optimized all neural models with the AdamW algorithm. The learning rate and the batch size were 1e-3 and 64, respectively. We applied early stopping based on the F1 score on the validation data. The maximum number of tweets per input was 200. For symptom and user prototypes, we set the number of prototypes per class to $m = 7$ and $k = 3$, respectively. We sampled normally around the base prototypes with $\sigma = 0.025$. We also used two-layer attention for Step 3. In section B, we cover the extensive ablation study and hyperparameter tuning.

Table 1: Performance comparison for depression detection. **ProtoDep (Tuned)** is a model variant with a modified loss function, while **ProtoDep (Avg. D1–D5)** represents the mean (±std) across five datasets.

| Method / Dataset | Precision (P) | Recall (R) | F1 Score |
|---|---|---|---|
| Gui et al. (2019) | 0.900 | 0.901 | 0.900 |
| Lin et al. (2020) | 0.903 | 0.870 | 0.886 |
| Zhang et al. (2021b) | 0.909 | 0.904 | 0.912 |
| Han et al. (2022) | 0.975 | 0.969 | 0.972 |
| **ProtoDep (Tuned)** | **0.985** | **0.995** | **0.990** |
| **ProtoDep (Avg. D1–D5)** | $0.934\pm0.036$ | $0.954\pm0.020$ | $0.944\pm0.026$ |
| *Granular Performance for ProtoDep* | | | |
| D1 | 0.964 | 0.953 | 0.959 |
| D2 | 0.898 | 0.951 | 0.924 |
| D3 | 0.984 | 0.991 | 0.987 |
| D4 | 0.931 | 0.931 | 0.931 |
| D5 | 0.893 | 0.946 | 0.919 |

**Result (1): Classification Performance.** ProtoDep achieved competitive results across five benchmark datasets, demonstrating high accuracy while providing interpretable classifications. As summarized in Table 1, ProtoDep achieved an average F1 score of 94.4%, maintaining consistent performance across the five randomly sampled datasets (D1-D5). Notably, the ProtoDep (Tuned) variant achieved a state-of-the-art F1 score of 99.0%, highlighting its effectiveness. However, a trade-off in interpretability was observed for ProtoDep (Tuned), which we address in section B.2.

We performed k-fold cross-validation (k=5) to validate robustness, confirming that the results were not driven by overfitting or data leakage. We also tested statistical significance across the compared models, with ProtoDep showing statistically significant improvements in F1 score over baseline models ($p < 0.05$).

**Result (2): Explainable Prototypes.** We evaluate the quality of ProtoDep's learned prototypes using two distinct methods. **Alignment with Clinical Lexicon.** We compare learned prototypes with domain expert-labeled ground truth using a specialized dictionary from Yazdavar et al. (2017). This lexicon contains an extensive collection of depression-related terms specifically associated with the nine symptom categories of the PHQ-9. These terms have undergone meticulous curation to capture the subtle nuances inherent in depression symptoms. By aligning our learned prototypes with this established lexicon, we can assess their relevance and meaningfulness within the context of depression. To quantify this alignment, we compute the mean representation for each symptom prototype and subsequently measure its cosine similarity with the embedded ground truth lexicon.

Figure 2 shows strong alignment between the learned symptom prototypes and clinically relevant depression terms, reflected in high diagonal self-similarity scores. Notably, 'Lack of Interest' and 'Feeling Down' exhibit elevated similarity with other symptoms, likely due to their overlap in clinical settings and initialization bias (Appendix B.7).

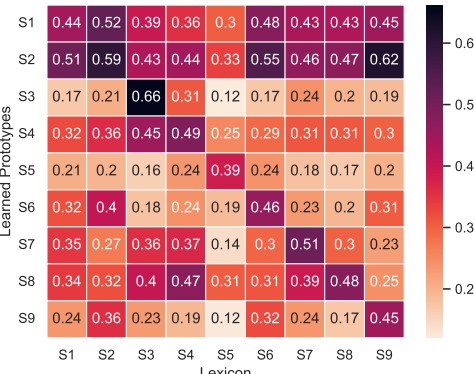

Figure 2: Similarity between learned symptom prototypes and ground truth lexicon.

**Discriminative Power with PRIDE Score.** We assessed the prototypes' discriminative power by using the PRIDE score Ni et al. (2022), inspired by Zhang et al. (2021a). This method defines a "real" prototype for each category by averaging its data points and then measures the similarity between these real prototypes and the learned ones. A high PRIDE score indicates a learned prototype's effectiveness in capturing its designated category while differentiating itself from others. We assume the nearest tweet to each symptom in the ground truth lexicon is the real prototype in the dataset. Figure 3 demonstrates that ProtoDep achieves positive PRIDE scores for all symptoms' prototypes, implying effective learning of distinct and representative symptom prototypes.

We also assessed the efficacy of learned user prototypes by reporting the PRIDE score. Specifically, we identify a representative 'real' user for each class by computing the mean of all users within that class and calculating the PRIDE score. Notably, both depressed and non-depressed classes exhibit positive PRIDE scores (0.27 and 0.33, respectively), affirming that ProtoDep effectively captures a meaningful prototype space for users.

**Result (3): Transparent Decision-Making.** Beyond accurate depression detection, ProtoDep offers valuable insights into its decision-making process through several avenues. Examining the weights assigned to various symptoms within its final layer unveils their relative importance in user classification. As illustrated in Figure 4, ProtoDep across diverse datasets prioritizes symptoms like "Fatigue or low energy" and "Lack of Interest," mirroring human expert judgment reported in Yazdavar et al. (2017). Interestingly, it assigns less weight to "Sleep Disorder" and "Concentration problems," potentially due to the ambiguity of these symptoms in textual data. For example, the tweet "lost in my own mind" might not explicitly mention keywords indicating "Concentration problems," making accurate classification challenging. This finding highlights the inherent difficulty in capturing nuanced depressive symptoms, even for human experts. Furthermore, ProtoDep's user embedding layer with stacked attention layers holds promise for interpreting user classifications, similar to Han et al. (2022). We analyzed attention scores to identify tweets that significantly influence user classification. However, echoing prior research Bibal et al. (2022); Wen et al. (2022); Pruthi et al. (2020), our extensive evaluation across both methods revealed no statistically significant association between attended tweets and those

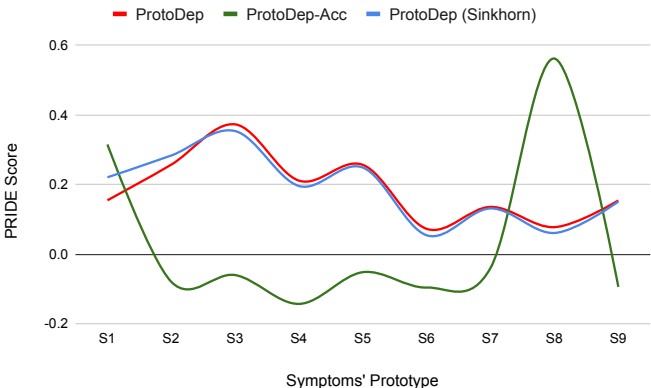

Figure 3: Visualization of the PRIDE score for learned prototypes for ProtoDep, ProtoDep (Tuned), and ProtoDep (Sinkhorn) Models.

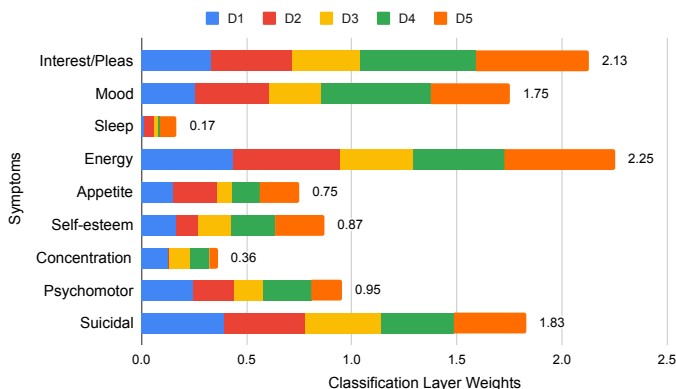

Figure 4: Classification weights (absolute values) for different symptoms over all datasets.

crucial for accurate classification. This suggests that while attention weights offer glimpses into model behavior, they might not directly explain specific classification outcomes in this context.

## 5. Conclusion

We introduced ProtoDep, a neurosymbolic framework that fuses PHQ-9 symptom categories with LLM-guided prototype learning to yield three-tier interpretability—symptom mappings, case-based analogies, and transparent decision weights. Across five benchmarks and ablations, ProtoDep achieves 94% F1 (and 99% for the Tuned variant) while PRIDE scores verify prototype fidelity, highlighting a performance–interpretability trade-off. This work lays groundwork for interpretable, AI-driven depression screening; future efforts will validate with clinical experts and extend to diverse social platforms.

## Acknowledgments

This research was funded by the National Science Foundation under Grant 2033521 A1, titled "KnowWhereGraph: Enriching and Linking Cross-Domain Knowledge Graphs using Spatially-Explicit AI Technologies." The views, findings, conclusions, or recommendations presented in this paper are those of the authors and do not necessarily reflect the views of the National Science Foundation.

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

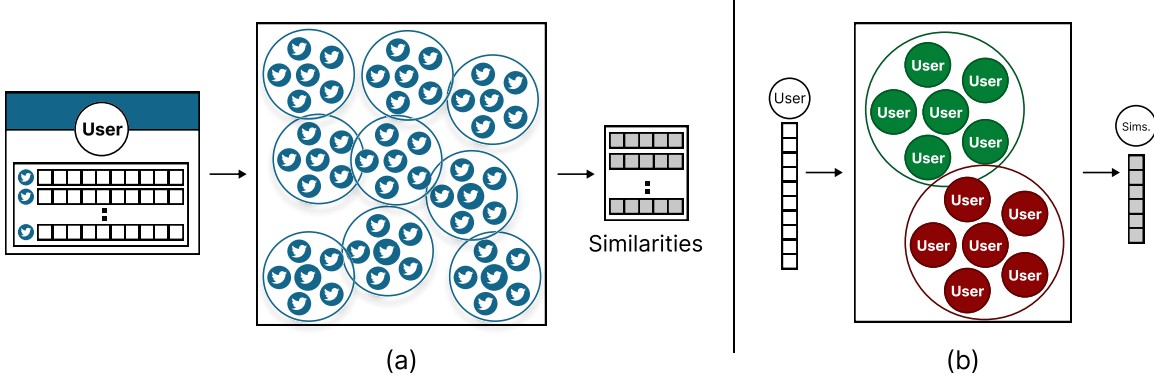

Figure 5: (a) Symptom Prototype Layer (b) User Prototype Layer

Table 2: Overview of five sampled datasets, showing the total number of positive and negative tweets. The datasets contain 2,159 positive users and 2,049 negative users in total. (Adapted from Han et al. (2022)).

| Dataset | Total Positive Tweets | Total Negative Tweets |
|---|---|---|
| D1 | 156,013 | 153,328 |
| D2 | 151,538 | 119,188 |
| D3 | 142,057 | 118,611 |
| D4 | 143,725 | 124,925 |
| D5 | 148,039 | 134,700 |

Baoquan Zhang, Xutao Li, Yunming Ye, Zhichao Huang, and Lisai Zhang. Prototype completion with primitive knowledge for few-shot learning. In *Proceedings of the IEEE/CVF Conference on Computer Vision and Pattern Recognition*, pages 3754–3762, 2021a.

Dongyu Zhang, Nan Shi, Ciyuan Peng, Abdul Aziz, Wenhong Zhao, and Feng Xia. Mam: A metaphor-based approach for mental illness detection. In *International Conference on Computational Science*, pages 570–583. Springer, 2021b.

## Appendix A. Dataset

We employ an openly available Twitter (now X) dataset, MDL, specifically designed for depression detection, using the version provided by Han et al. (2022). In this dataset, users who posted tweets containing predefined phrases indicative of depression—such as "I'm depressed" or "I've been diagnosed with depression"—were labeled as depressive. Those who never posted any tweet containing the substring "depress," after contextual filtering (e.g., excluding instances like "depressed of joy"), were labeled as non-depressive. As clinician-verified ground truth falls outside the scope of this work, we instead use existing silver standard labels alongside expert-curated PHQ-9 lexicons for evaluation. Han et al. (2022) created five distinct splits via random user selection: 60% train, 20% validation, and 20% test, yielding 2,524 users for training and 842 users each for validation and testing (Table

Table 3: Comparison of average validation F1 scores for different initialization methods compared to different loss functions. "MSE." refers to Mean Squared Error loss, and "Ent." refers to Entropy loss.

| Loss Function | GPT-4 | Lexicon | Lex. Tweets |
|---|---|---|---|
| Triplet+MSE.+Ent. | 0.990 | 0.990 | 0.989 |
| Sinkhorn+MSE. | 0.947 | 0.969 | 0.954 |
| Sinkhorn | 0.936 | 0.964 | 0.949 |
| Avg. val. F1 | 0.958 | **0.975** | 0.964 |

2). Although our MDL splits are balanced, the real-world prevalence of depressive signals is much lower; future work should evaluate skewed class distributions or incorporate cost-sensitive training. While this dataset originates from X, ProtoDep is platform-agnostic; any social media feed that is amenable to sentence embeddings can integrate seamlessly into our pipeline.

## Appendix B. Ablation Study

We consider four different settings to validate the impact of different hyperparameters.

### B.1. Symptom Prototype Initialization.

In this study, we explore alternative methods for initializing symptom prototypes. We compare two novel initialization approaches with the baseline method that utilizes a pre-trained language model (LLM) for symptom initialization. In the first setting, we leverage the ground truth lexicon as the foundation for symptom prototypes. We extract ground truth embeddings from this lexicon and then sample additional prototypes around them for each symptom class. A well-constructed lexicon captures domain-specific nuances and expert knowledge, which can enhance the quality of symptom prototypes. In the second setting, we depart from direct lexicon embeddings. Instead, we identify the nearest tweet in our dataset to each lexicon symptom and use its embedding as the basis for symptom prototypes. We then continue sampling around these tweet-grounded prototypes. By anchoring the initial prototypes to actual tweets, we aim to improve their relevance and alignment with real-world symptom expressions. Our experimental results in Table 3 demonstrate that both the lexicon-based and tweet-grounded initialization outperform the LLM baseline. Notably, the curated lexicon's consideration of various combinations of the depression-indicative keywords contributes to its effectiveness. However, the marginal performance difference between these two settings and the baseline suggests that LLMs can achieve competitive results even in scenarios lacking human annotation or domain-specific knowledge.

### B.2. Prototype Loss Function.

To assess the influence of different loss functions within the ProtoDep framework, we implemented two evaluation settings. The first setting, ProtoDep (Sinkhorn), exclusively employed the Sinkhorn loss to isolate the impact of the MSE loss in ProtoDep. The second

Table 4: Comparison of different attention mechanisms.

| Attention | Baseline | Multi-head |
|---|---|---|
| Avg. val. F1 | **0.9407** | 0.823 |

setting, ProtoDep (Tuned), combined Triplet loss with MSE and Entropy losses, mimicking conventional loss functions commonly used in prototype learning research.

As observed in Table 3, ProtoDep (Tuned) achieved strong performance, demonstrating the capability of the ProtoDep framework. Notably, it surpassed other settings in terms of F1 scores. However, as illustrated in Table 3, while ProtoDep (Tuned) yielded higher F1 scores, its PRIDE scores indicated a failure to learn meaningful symptom-level prototypes. Conversely, ProtoDep (Sinkhorn) achieved better PRIDE scores, signifying successful prototype learning, but yielded lower F1 scores than ProtoDep.

This evaluation highlights a trade-off between classification performance and interpretability within the ProtoDep framework. While ProtoDep (Tuned) excelled in F1 scores, its learned prototypes lacked interpretability. In contrast, ProtoDep and ProtoDep (Sinkhorn) prioritized interpretability through meaningful prototypes but compromised classification accuracy. These findings suggest the need for careful consideration of loss function selection in balancing interpretability and performance within the ProtoDep framework.

### B.3. User Embedding Model

We introduce a single-layer multi-head attention configuration instead of employing the conventional user embedding attention mechanism. The outcomes are presented in Table 4, which indicates the effectiveness of ProtoDep.

### B.4. Number of Prototypes

Figure 6 depicts the average F1 score across all five datasets, varying the number of prototypes. While the overall trend suggests that the model performs better with fewer prototypes, a nuanced examination reveals that individual datasets often favor a larger number of prototypes.

### B.5. Tweet Embedding Model

We chose "all-mpnet-base-v2" Reimers and Gurevych (2019) as our Tweet Encoder Model due to its strong capability to capture semantic similarity across generalized contexts, which is particularly useful for varied language use on social media. We found fine-tuned embedding models, such as BERTweet Nguyen et al. (2020), do not provide significant contributions in this domain.

### B.6. Sample Symptom Prototypes

To demonstrate the difference between the quality of learned prototypes and base prototypes for different initialization techniques, we present five different tweets. Each tweet expresses a different symptom or emotion, and Figures 7, 8, 9, 10, 11 illustrate their similarity to the prototypes. Based on our experiment, we observe that the prototypes that were initialized

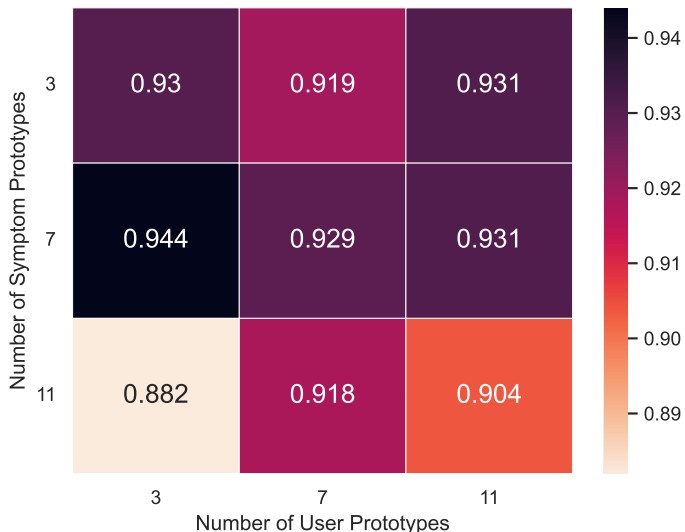

Figure 6: Average F1 score over all test datasets for different numbers of prototypes.

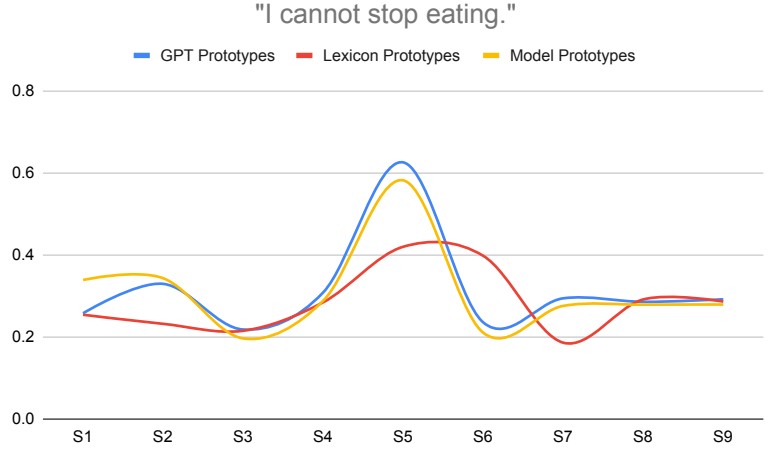

Figure 7: Cosine similarity of the tweet at the top with different prototypes.

with GPT-4 (including ProtoDep learned prototypes) explain the latent symptoms of the tweets more intuitively than the ones that were initialized by our ground truth Lexicon.

We also analyzed the closest training examples for each of our 45 symptom prototypes, as shown in Table 5. Examining the table reveals that the symptom prototypes are meaningful and effectively capture the different aspects of a symptom. For instance, the learned prototypes for Changes in appetite symptoms, such as "remember, hungry is skinny," "Flat stomach and tiny legs thinspo," and "I'm a fat mess," reflect the contrasting signs of this symptom. This demonstrates that the Proto-Dep symptom layer successfully learned mean-

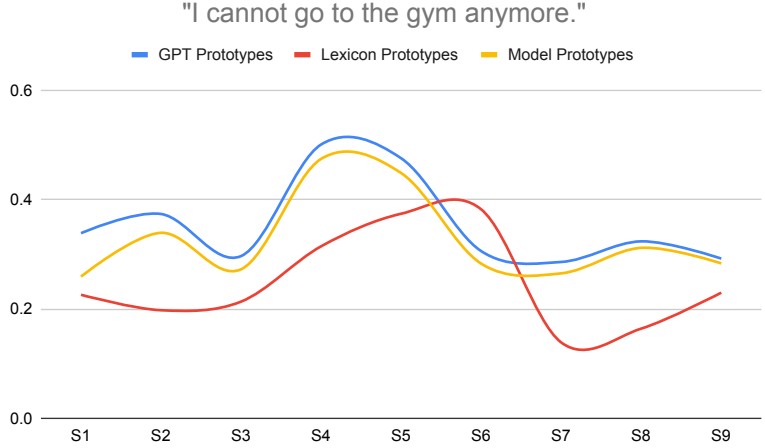

Figure 8: Cosine similarity of the tweet at the top with different prototypes.

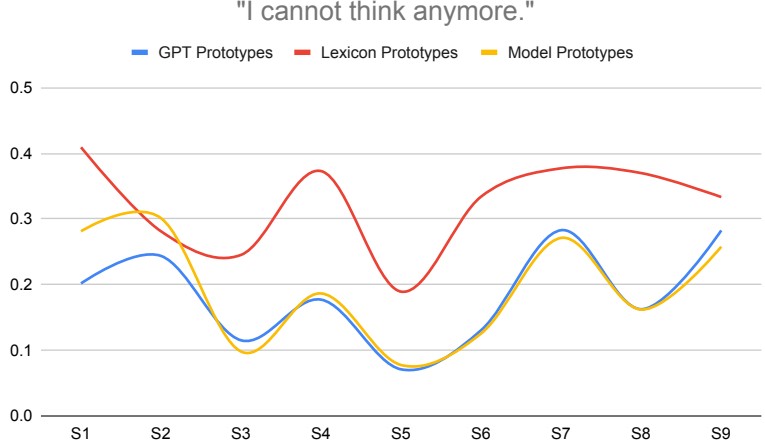

Figure 9: Cosine similarity of the tweet at the top with different prototypes.

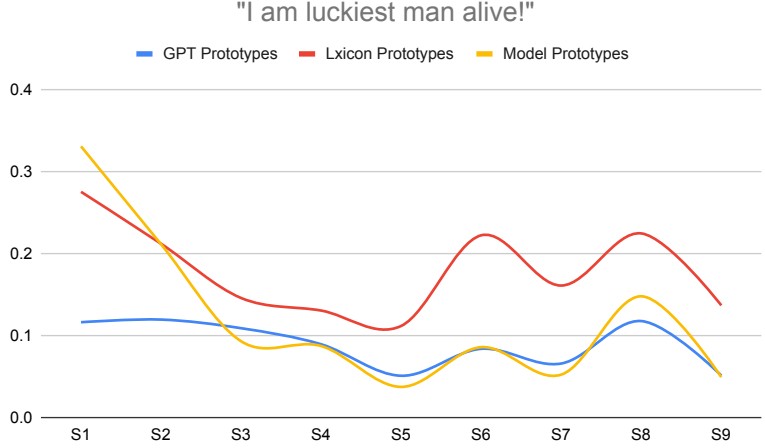

Figure 10: Cosine similarity of the tweet at the top with different prototypes.

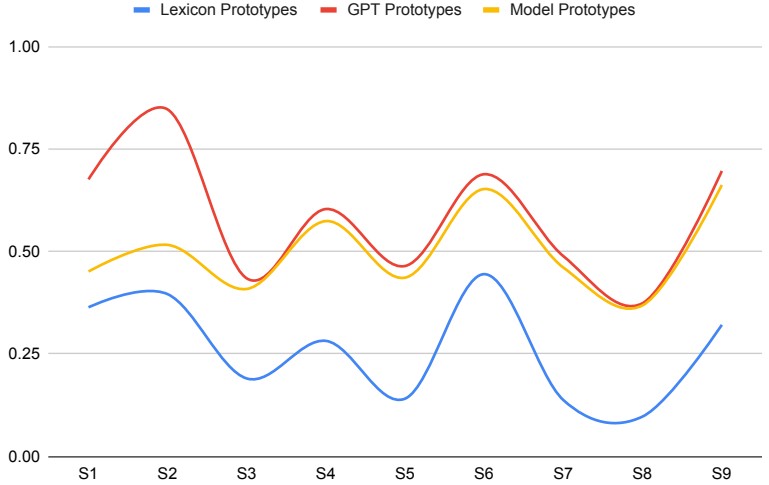

Figure 11: Cosine similarity of the tweet "I don't know how to cope anymore. I feel hopeless, worthless, and guilty all the time. I can't sleep, eat, or concentrate. I have no interest in anything. I wish I could just disappear. depression" which represents many symptoms with different prototypes.

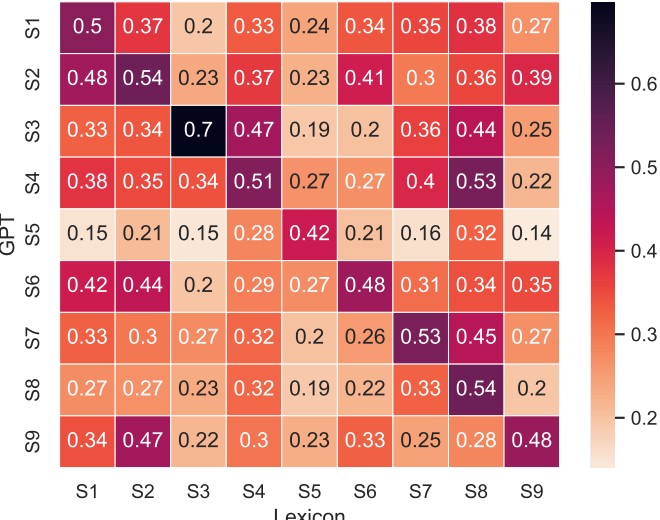

Figure 12: Similarity between initial GPT prototypes and ground truth lexicon.

ingful representations for each symptom. However, some of the retrieved prototypes appear meaningless. This could indicate either the absence of a suitable example in the dataset to capture the prototype's true meaning or the inherent complexity or simplicity of the symptom, potentially requiring a different number of prototypes.

## B.7. Symptom Space Creation

The similarity between initial GPT prototypes and ground truth lexicon is illustrated in Figure 12.

## B.8. GPT-4 Prompts

We provide the prompt to develop the initial prototype of symptoms illustrated in Figure 13. Additionally, we illustrate a sample response in Figure 14.

| Symptoms | Nearest Tweets to the Prototypes |
|---|---|
| Depressed Mood | I am not happy.
I'm having a bad night
drowning in an ocean of tears
I am not happy .
Sad and annoyed _()_/ |
| Loss of interest or Pleasure | How I feel daily.
people that texts slow.
I'm having a bad night
is already looking brighter
drowning in an ocean of tears |
| Sleep Disturbance | what's got you awake so early
tired just woke up cant go back to sleep
Too much weed n drugs maybe
Sleep Schedule = Shambles
Ready for a long nap |
| Fatigue or Low Energy | Why thats so hard i be waisting the shit out my time '
team follow back Editor's note: this post contains an image ...
Tired , everything above
Tired is an understatement.
tired of being tired |
| Changes in Appetite | Remember, hungry is skinny.
Somebody that's not normal
Flat stomach and tiny legs thinspo
I'm a fat mess
that is very fat |
| Feelings of Guilt or Worthlessness | Noooo babe feel better
I feel like a failure
I'm waste, I'll end this there
I'm miserable I'm disgusting
but i can't sleep |
| Difficulty Concentrating | lost in my own mind
you spelled remember wrong
undateable life wants YOU ... If u don't laugh, you'll get ur back...
My memory is failing me today
ur question is ______ *whisper* |
| Psychomotor Agitation or Retardation | Psychomotor agitation or retardation
Ano nood mo beby
my job isn't that bad tho. it's just tiring.
I'm getting too careless.
WHAT IS GOING ON
WHAT KIND OF QUESTION IS THIS |
| Suicidal Thoughts | What Dying For Seconds Taught Me About Living -
When I think of my son, I feel love and pure joy. That wasn't a...
UN spox says complete meltdown of humanity in *** amid reports of massacres...
Do you wanna die
—— MY HEART IS HURTING
If ever follows me |

Table 5: Nearest tweets from the dataset to ProtoDep's Symptom Prototypes.

You are tasked to generate a comprehensive dataset of tweets that express depressive symptoms. The data you generate will serve as an initial dataset for machine learning research about depression detection on social media. The data will not be used in any way by human participants. To do this task, you have first to consider all the PhQ-9 symptoms of depression. Second, you must list the different aspects of each symptom that can be expressed. Third, for each symptom aspect, write enough comprehensive tweets to express that aspect. Give at least two examples for each aspect. Generate the last step in a CSV format.

Follow this format:

PHQ-9 Symptom: Symptom_name
Aspect 1: Aspect description
Tweet 1: Tweet text
Tweet 2: Tweet text
Aspect 2: Aspect description
Tweet 1: Tweet text
Tweet 2: Tweet text
…

PHQ-9 Symptom: Symptom_name
Aspect 1: Aspect description
Tweet 1: Tweet text
Tweet 2: Tweet text
…

Figure 13: The prompt to GPT-4 to generate the initial symptom examples.

That sounds like an interesting task. I will try to create a comprehensive sample set for each Patient Health Questionnaire-9 (PHQ-9) symptom, using sentences that could appear in social media. Here is what I came up with:

1. Little interest or pleasure in doing things
    - I used to love painting, but now I can't even pick up a brush. It feels like nothing matters anymore.
    - Nothing excites me these days. I just scroll through my feed, feeling numb and bored.
    - I don't enjoy anything I do. I just go through the motions, hoping for something to change.
    - I have no motivation to pursue my goals. I've lost interest in everything that used to make me happy.
    - I feel like I'm wasting my life. I don't have any hobbies, passions, or joys. I just exist.

2. Feeling down, depressed, or hopeless
    - I feel so low and hopeless. I don't see any point in living. I just want to end this pain.
    - I'm drowning in sadness. I can't breathe. I can't smile. I can't cope.
    - I'm trapped in a dark hole. I can't climb out. I can't see any light. I can't find any hope.
    - I'm broken and worthless. I don't deserve anything good. I don't belong anywhere.
    - I'm tired of fighting. I don't have any strength left. I don't have any reason to live.
….

Figure 14: A sample response to the prompt by GPT-4.

## Appendix C.  Ethical Consideration

We used a publicly available dataset introduced by Han et al. (2022). Our investigation focuses exclusively on textual content, deliberately excluding user profile information. We emphasize the ethical implications of our model's application and strongly discourage misuse compromising data security or privacy principles.

## Appendix D.  Limitations

While ProtoDep offers a promising approach to interpretable depression screening on social media, we note the following limitations:

**Gold-Standard Clinical Annotations:** While direct clinician annotation falls outside this study's scope, we approximate prototype faithfulness using expert-curated PHQ-9 symptom lexicons Yazdavar et al. (2017) and PRIDE analyses. Validation in collaboration with healthcare professionals is planned for future work.

**Data Representativeness:** The learned prototypes reflect biases present in the MDL dataset; underrepresented demographics or groups may receive less accurate symptom mappings.

**Demographic Generalizability:** Our dataset lacks metadata on age, gender, and cultural background. Assessing ProtoDep's performance across diverse subpopulations is critical for equitable deployment.

**Scalability:** Case-based reasoning requires nearest-neighbor lookups over tweet and user prototypes; scaling to millions of users will necessitate approximate indexing (e.g., FAISS) to maintain efficiency.

**Tweet Encoder Dependency:** ProtoDep's prototype learning depends on the expressiveness of the underlying sentence encoder. Poor embeddings may inhibit convergence to meaningful prototype clusters.

**Privacy Concerns:** Leveraging social media posts for mental health inference raises inherent privacy issues. Strict adherence to ethical and legal guidelines is required in any downstream application.

**Platform Scope:** While evaluated on Twitter (now X), ProtoDep is embedding-agnostic and can be applied to other social feeds; however, performance on platforms with different linguistic or behavioral norms remains to be validated.

**Hyperparameter Sensitivity:** Prototype quality and classification performance vary with hyperparameter choices; careful tuning is necessary when adapting ProtoDep to new domains.

