# OpenReview forum: "Towards Explainable Depression Detection: A Neurosymbolic Approach to Uncover Social Media Signals with Generative AI"
_nesyconf.org/NeSy/2025/Conference — NeSy 2025 Poster_

### Official Review · Reviewer_ndQp · 2025-04-04
**Towards Explainable Depression Detection: A Neurosymbolic Approach to Uncover Social Media Signals with Generative AI**

**Rating:** 7
**Confidence:** 4

**Review:**

The article reports a pipeline to use language model embedding-based similarity metrics to define prototypes for depression symptoms that can be applied  to similarly embedded social media posts (tested on a Twitter-extracted annotated dataset) to detect depression symptoms. In general I like the method and I think it brings novel elements. However I have several concerns.

1. In the conclusion it is stated that 'Additionally, ProtoDep (tuned) achieves 99% ac-
curacy, demonstrating that our framework can match state-of-the-art performance while
remaining interpretable.'. However, in the Appendix it is stated that 'However, as illustrated in Table 3, while ProtoDep (Tuned) yielded higher F1 scores, its PRIDE scores indicated a failure to learn meaningful symptom-level prototypes.
Conversely, ProtoDep (Sinkhorn) achieved better PRIDE scores, signifying successful pro-
totype learning, but yielded lower F1 scores than ProtoDep.'
Taken together, I think the first claim is misleading - the authors themselves state clearly in the appendix that *either* the system achieves state of the art performance *or* is interpretable (as measured by PRIDE score) but not both.

2. Use of Twitter: I believe that Twitter data is no longer made available for research (nor is Twitter even called Twitter any longer) thus I wonder if the authors might consider showing their method works on a more recent data source. The generalisability of a historical Twitter dataset seems uncertain to me as social media platforms evolve very rapidly.

3. Concern with the labelling used to evaluate the algorithm: The paper states that the labelling in the dataset that they used was performed as follows: 'In this dataset, individuals who posted tweets
containing predefined phrases indicative of depression, such as ”I’m,” ”I was,” ”I am,” or
”I’ve been diagnosed with depression,” were labeled as depressive. Conversely, those who
never posted any tweet containing the term ”depress” were labeled as non-depressive.'
It may be only my lack of understanding but it seems to me that it would be trivial to automatically detect mentions of "depress" in a text, thus, if those texts are still included in the dataset it would be trivial to reproduce the annotations 100%. Moreover, it seems that aggregating to individuals this way  is at best a very approximate connection to the actual content of the other messages that an individual may post on a social media platform. Could the authors better explain why the challenge dataset is not trivial?

**Anonymity:**

Remain anonymous

---

### Official Review · Reviewer_fKpZ · 2025-04-05
**Nice application of Neuro-Symbolic AI**

**Rating:** 6
**Confidence:** 4

**Review:**

The paper proposes ProtoDep, a neuro-symbolic method that is designed to detect depression through twitter data while improving interpretability by providing explanations at three levels. The patient health questionnaire-9 (PHQ-9) is used as a reference to define the concept while GPT-4 is used to generate examples. The experiments on the MDL data set are interesting and show that ProtoDep is effective in classification and explanation of prototypes.

Strengths:

1. The application is very interesting and a neuro-symbolic approach is both natural and ideal in these sort of scenarios. There is dearth of real-world application in NSAI and this work plugs the gap a little bit.

2. The aggregation of tweet-level information into user-level representations, although straightforward, is important because mental health signals may be spread across multiple social media posts and this representation makes it simpler to identify this.

3. The explanability aspect is also nice with the similarity between learned prototypes and the lexicons in PHBO-9 as shown in figure 2 nicely demonstrates ProtoDep's ability to be useful to end users.

Weaknesses:

1. I am missing a human aspect here. The model is expalainable but asking a real human expert to corroborate the findings would have been a strong point for this work. I am aware that the authors cite Yazdavar et al. (2017) to justify the alignment with human expert but the assocviated figure does not bring this out. Having a human expert without relying on previous works is ideal.

2. There's little discussion on how and if themodel generalizes across different aspects or subpopulations such as age, gender and  culture of the person as this is especially critical for sensitive applications like mental health.

3.  Case based reasoning does not scale well and thus scalability of this system to tens of millions of tyweets seems to an issue.

4. The used data set is relatively balanced but the reality, in my opinion, is completely different as there will far less positive examples (tweets signifying issues). Such a scenario should also be considered.

5. In symptom space optimization, using a similraity measure is fine but not a very principled choice since these measures can spring up surprises with the labeling of the tweet i.e. the symptom not making sense. A discussion on this will be appreciated.

6. Minor: Figure 1 caption needs to be more detailed. Just writing "Overview" is not enough.

Overall. although the paper has several issues but there are nice ideas in there and the problem it is tackling is important and thus I am leaning towards towards acceptance.

**Anonymity:**

Remain anonymous

---

### Official Review · Reviewer_7vwQ · 2025-04-07
**Interesting problem, results seem promising, but are they clinically relevant?**

**Rating:** 6
**Confidence:** 3

**Review:**

The paper introduces ProtoDep, a neurosymbolic framework designed for depression detection using social media data. It combines PHQ-9 symptom categorizations with neural prototype learning, guided by large language models, to provide multi-level explanations. The authors report a 94% F1-score across five datasets, suggesting potential for real-world applications.


Pros:

1. The integration of symbolic medical constructs with neural embeddings is a significant advancement in making AI models more interpretable, especially in sensitive domains like mental health. Personally, I am glad to see this topic being researched and I thank the authors for their effort.

2. High F1-score across multiple datasets.

3. The multi-level explainability via symptoms, user profiles and even the lexicon used. These signals are clear and quantifiable which are crucial for clinical adoption and trustworthiness in mental health assessments.



Cons:

1. A significant limitation of this project is the lack of gold data. The authors attempt to address this by generating synthetic, silver data for symptoms and user labels, which in itself is quite elegant. While the evaluation suggests that this approach is successful, the clinical value of such synthetic data remains questionable. The lack of robust, clinically relevant gold data could undermine the reliability of the findings.
While the framework shows promise, its effectiveness in real-world clinical settings requires further validation through controlled studies with healthcare providers.

2. The reliance on Twitter data raises concerns about the completeness and representativeness of the user data. Tweets may not fully capture a user's mental state, leading to skewed or incomplete assessments. The approach's generalizability to other platforms or real-world settings is uncertain, given the dependency on the underlying gold dataset's validity.

3. Same goes for explainability: the paper offers explainability by comparing user tweets with prototypical profiles and symptom prototypes. However, the clinical reliability and value of this approach are unclear.
While the paper emphasizes explainability, it does not convincingly demonstrate how the explanations provided by ProtoDep are actionable or useful in clinical settings. The reliance on prototype-based reasoning is not sufficiently validated or explained.


While the paper presents an interesting approach to enhancing explainability in depression detection through social media, it is significantly undermined by several weaknesses. The lack of clinical validation and limited data scope are major concerns.
Despite claims of real-world applicability, the paper does not provide any clinical validation or evidence that ProtoDep's outputs are useful or actionable for mental health professionals. Without such validation, the framework's practical utility remains speculative.
The submission might be more appropriate for a domain-specific venue, and it requires more extensive clinical validation to establish its utility in practical settings.

**Anonymity:**

Remain anonymous